# Impact of ABC Transporters in Osteosarcoma and Ewing’s Sarcoma: Which Are Involved in Chemoresistance and Which Are Not?

**DOI:** 10.3390/cells10092461

**Published:** 2021-09-17

**Authors:** Massimo Serra, Claudia Maria Hattinger, Michela Pasello, Chiara Casotti, Leonardo Fantoni, Chiara Riganti, Maria Cristina Manara

**Affiliations:** 1Laboratory of Experimental Oncology, IRCCS Istituto Ortopedico Rizzoli, Via di Barbiano 1/10, 40136 Bologna, Italy; claudia.hattinger@ior.it (C.M.H.); michela.pasello@ior.it (M.P.); chiara.casotti@ior.it (C.C.); leonardo.fantoni@ior.it (L.F.); mariacristina.manara@ior.it (M.C.M.); 2Department of Oncology, University of Torino, Via Santena 5/bis, 10126 Torino, Italy; chiara.riganti@unito.it

**Keywords:** osteosarcoma, Ewing’s sarcoma, ABC transporters, drug resistance, pharmacogenomics, tailored treatment

## Abstract

The ATP-binding cassette (ABC) transporter superfamily consists of several proteins with a wide repertoire of functions. Under physiological conditions, ABC transporters are involved in cellular trafficking of hormones, lipids, ions, xenobiotics, and several other molecules, including a broad spectrum of chemical substrates and chemotherapeutic drugs. In cancers, ABC transporters have been intensely studied over the past decades, mostly for their involvement in the multidrug resistance (MDR) phenotype. This review provides an overview of ABC transporters, both related and unrelated to MDR, which have been studied in osteosarcoma and Ewing’s sarcoma. Since different backbone drugs used in first-line or rescue chemotherapy for these two rare bone sarcomas are substrates of ABC transporters, this review particularly focused on studies that have provided findings that have been either translated to clinical practice or have indicated new candidate therapeutic targets; however, findings obtained from ABC transporters that were not directly involved in drug resistance were also discussed, in order to provide a more complete overview of the biological impacts of these molecules in osteosarcoma and Ewing’s sarcoma. Finally, therapeutic strategies and agents aimed to circumvent ABC-mediated chemoresistance were discussed to provide future perspectives about possible treatment improvements of these neoplasms.

## 1. Introduction

The ATP-binding cassette (ABC) transporter superfamily consists of several proteins with a wide repertoire of functions, including the transport of a broad spectrum of chemical substrates and chemotherapeutic drugs.

So far, 49 human genes and 21 pseudogenes have been identified for ABC proteins, which are conventionally grouped into seven families (named with letters from A to G) according to their relative sequence homology and domain organization [1], although a new ABC classification system has been recently proposed [2].

Under physiological conditions, ABC transporters are involved in cellular trafficking of hormones, lipids, ions, xenobiotics, and several other molecules, as well as in intracellular regulation of organelles such as mitochondria, lysosomes, the endoplasmic reticulum, or the Golgi apparatus [3].

In cancers, ABC transporters have intensively been studied in the past decades, mostly because of their involvement in the multidrug resistance (MDR) phenotype [4]. Indeed, the most common mechanism leading to MDR in human tumors is the increased activity or overexpression of ABC proteins that reduce the levels of chemotherapeutic drugs accumulation within cells [5].

Osteosarcoma (OS) and Ewing’s sarcoma (ES) are, together with chondrosarcoma, the most common primary tumors of bone, although they are rare neoplasms [6]. Studies on ABC transporters in these tumors have provided findings that have partially been translated to clinical practice.

Figure 1 schematically represents those ABC transporters, which are overviewed in this review in relation to their expression and activity in OS and ES tumor cells, as well as in their cancer stem cells.

ABC functions, both related and unrelated to drug resistance, are also discussed.

Finally, potential therapeutic strategies and novel agents used to circumvent the ABC-mediated chemoresistance or to sensitize bone tumor cells to conventional chemotherapeutic drugs are also reviewed.

## 2. ABC Transporters and Drug Resistance in Osteosarcoma

The different backbone drugs used in first-line or rescue chemotherapy for high-grade OS are substrates of ABC transporters, which can confer drug resistance through increased efflux of chemotherapeutic agents [7,8,9].

Several experimental studies have demonstrated that human OS cell lines can become chemoresistant by overexpressing ABC proteins, since the presence of increased levels of these transporters on the cell membrane enhances the efflux of anticancer drugs. For example, increased expression of ATP-binding cassette subfamily B member 1 (*ABCB1*; also known as *MDR1* or P-glycoprotein) was found to be associated with the degree of doxorubicin resistance [10] and to unresponsiveness to other drugs that are substrates of this transporter and are used in first-line or rescue OS treatments [11].

Increased *ABCB1* levels have been also proven to be associated with resistance to paclitaxel in human OS cell lines, which developed cross-resistance to other *ABCB1* substrates such as doxorubicin, docetaxel, and vincristine [11]. In another study, doxorubicin-resistant human OS cell lines with increased *ABCB1* expression were found to be cross-resistant to taxotere, etoposide, and vinorelbine [12]. On the other hand, the causal involvement of *ABCB1* overexpression for doxorubicin resistance in human OS cells was demonstrated by studies proving that decreasing or knocking down *ABCB1* expression resulted in restoration of doxorubicin sensitivity [12,13].

This body of evidence has clearly indicated that in OS, overexpression of ABCB1 can determine resistance to different drugs that are used not only in first-line chemotherapy protocols, but also in rescue treatments.

By considering the ABC transporters that have most consistently been associated with the degree of resistance against doxorubicin, methotrexate, or cisplatin in human tumors and experimental models [14,15,16,17], a group of 12 ABC transporters was studied in a panel of 6 drug-sensitive and 20 resistant variants of the U-2OS and Saos-2 human OS cell lines [12]. *ABCB1* and ATP-binding cassette subfamily C member 1 (*ABCC1*) emerged as the most relevant transporters associated with the degree of drug resistance in OS cells, being overexpressed in resistant variants in comparison with parental cells; however, it is interesting to note that *ABCB1* protein levels were higher in all doxorubicin-resistant variants in comparison with both parental cells and other drug-resistant cell lines, confirming its specific involvement in resistance against this drug. In contrast to *ABCB1*, *ABCC1* protein expression was increased in all resistant variants and was not associated with resistance to any specific drug [12].

These results further supported the relevance of these two ABC transporters, in particular for *ABCB1*, in the resistance against the backbone drugs used in OS chemotherapy.

In the past ten years, evidence showing that *ABCB1* and *ABCC1* are not the only ABC transporters involved in OS chemoresistance has also been reported. The ATP-binding cassette subfamily C member 4 (*ABCC4*) was indicated to be involved in doxorubicin resistance, since it was found to be increased in doxorubicin-resistant MG-63 cells and in primary cultures derived from a patient unresponsive to this drug [18]. Similarly, the ATP-binding cassette subfamily C member 5 (*ABCC5*) was found to be overexpressed in OS samples of patients with worse event-free survival, suggesting a possible role in poor drug response or drug resistance [19]. Finally, higher expression of the ATP-binding cassette subfamily G member 2 (*ABCG2*) was reported in drug-resistant OS cells compared to non-resistant ones [20].

It has to be underlined that these findings need to be further confirmed and expanded in order to evaluate the actual relevance of these ABC transporters for OS drug resistance and treatment response.

Another aspect that has to be considered is the interaction between OS and microenvironment cells, which is crucial in the control of OS development and in the progression and generation of immune-tolerant and immune-suppressive environments [21]. The mutual interactions with mesenchymal stem cells or between different clones of OS cells present within the tumor likely also contribute to drug resistance, thanks to the multiple exchanges of RNAs, proteins, and metabolites via extracellular vesicles [22]. Among the transferred molecules, *ABCB1* mRNA and protein have been abundantly detected in exosomes released from doxorubicin-resistant MG-63 cells and transferred to sensitive cells [23], suggesting that the horizontal transfer of *ABCB1* may be an important element determining the acquisition and maintenance of resistance. While this mechanism has been described to take place between sensitive and resistant OS cells, it is still not known whether it can also occur between stroma or immune-infiltrating and OS cells. If the latter is confirmed, it will mean that multiple circuitries determining *ABCB1*-mediated drug resistance characterize the OS microenvironment. On the other hand, a better knowledge of this cross-talk may pave the way for new strategies disrupting these circuitries and reversing drug resistance.

Unresponsiveness to first-line drug treatment is unfortunately a frequently encountered problem in the management of high-grade OS, being present in 40–50% of patients [24]. Different studies have demonstrated that this behavior can be associated with the overexpression of *ABCB1* at clinical onset [25,26,27,28,29,30], confirming the preclinical findings described above in clinical settings.

Although the clinical impact of *ABCB1* in high-grade OS has been debated [31,32], two meta-analyses [13,33] have confirmed its value in predicting clinical outcomes, and a similar body of evidence has been reported for other pediatric solid tumors [34]. These findings justify the efforts that have been made to target or inhibit *ABCB1* in OS, which are discussed below in Section 8.

*ABCB1*-increased expression has also been shown to be more frequent in pretreated high-grade OS compared with untreated tumors and to be linked to a higher degree of metastasis formation, although statistical significance for these correlations was not reached [35]. Moreover, metachronous lung metastases were reported to be more frequently *ABCB1*-positive compared with primary high-grade OS tumor samples, suggesting a gain of *ABCB1* expression in secondary lesions [36,37].

This body of evidence has led to the below considerations, which have relevant clinical impacts.

The increased *ABCB1* expression level that was detected in pretreated compared with untreated tumors [35] and in metachronous lung metastases compared with primary tumors [36,37] indicated that enhancement of this transporters can be induced by drug treatment, in addition to being a possible inherent feature of the tumor.

On the other hand, the higher frequency of *ABCB1* positivity in metachronous lung metastases [36,37] and the trend toward a higher degree of metastasis formation in patients with increased *ABCB1* levels [35] may indicate that *ABCB1* overexpression can also have a pro-metastatic effect; however, it was demonstrated that *ABCB1*-overexpressing human OS cells had decreased tumorigenicity and metastatic ability in athymic mice, together with a reduced migratory and invasive ability and a lower homotypic adhesion ability in vitro, indicating that *ABCB1*-increased expression levels were not associated with a greater malignant phenotype [38]. Accordingly, the incidence of *ABCB1* overexpression was found to be higher in OS patients with localized disease at the clinical onset than in those with evidence of metastasis at the time of diagnosis [38]. Taken together, these data indicated that both in vitro and in clinical settings, *ABCB1* overexpression is not associated with an increased aggressiveness of OS cells. Indeed, the correlation of *ABCB1* increased levels with worse treatment response and outcome was not related to a higher metastatic ability of *ABCB1*-overexpressing cells, but rather to their lack of responsiveness to cytotoxic drugs. Metachronous lung metastasis are, therefore, more frequently *ABCB1*-positive because they arise from OS cells that have survived the chemotherapy treatment due to their drug-resistant phenotype.

All of these considerations are of particular relevance, especially for patients who have developed drug resistance through *ABCB1*-related mechanisms, with obvious negative consequences for their clinical outcome and treatment probability.

## 3. ABCA Transporters: A Polyhedric Family of Transporters in Osteosarcoma

Not all ABC transporters that have been studied in OS have been proven to be directly associated with drug resistance. Recently, members of the ABCA family, most of which are physiologically linked to cholesterol and phospholipid transport, have emerged as new potential actors in OS progression [39]. Despite ATP-binding cassette subfamily A member 3 (*ABCA3*), ATP-binding cassette subfamily A member 5 (*ABCA5*), ATP-binding cassette subfamily A member 7 (*ABCA7*), and ATP-binding cassette subfamily A member 9 (*ABCA9*) upregulation having been found in OS [39], in most cases their functions are still unknown. Notably, *ABCA3* and *ABCA5* have been associated with chemoresistance, being present in a “side-population” enriched of cancer stem cells (CSCs) that are more resistant to chemotherapy [40,41]. Although *ABCA5* characterizes OS CSCs, together with *ABCG2*, it is not clear which transporters mostly contribute to the chemoresistance of this side-population [41]. On the other hand, high-throughput screening aimed at identifying drugs selectively killing OS CSCs identified 5-azacytidine, vincristine, and the antitelomerase RHPS4 as the most active agents [41], likely because they evade the transport activity of *ABCA5* and *ABCG2*. Besides chemoresistance, *ABCA5* also confers a highly pro-metastatic phenotype to OS CSCs [41], indicating that it should further be investigated as a potential negative prognostic marker in OS patients.

Recently, an interesting reciprocal correlation between the expression of *ABCB1* and of the ATP-binding cassette subfamily A member 1 (*ABCA1*), the main transporter of cholesterol, has been reported [42]. While *ABCB1* increased along with the degree of doxorubicin resistance in preclinical models of U-2OS cells, *ABCA1* was shown to be progressively downregulated. This ratio was proven to derive from the activation of *Ras*, which was dependent on the endogenous rate of synthesis of farnesyl pyrophosphate (FPP), an upstream metabolite of cholesterol. Indeed, FPP activates the *Ras*–extracellular signal-regulated kinase 1/2 (*ERK1/2*)–hypoxia-inducible factor-1α (*HIF-1α*) axis, which is a transcriptional inducer of *ABCB1*, as well as the Ras–Akt–mammalian target of rapamycin (*mTOR*) axis, which represses the *ABCA1* transcription mediated by the liver X receptor α (*LXRα*) [42].

Notably, *ABCA1* also effluxes isopentenyl pyrophosphate (IPP), a precursor of FPP and cholesterol [43]. IPP is the strongest endogenous activator of Vγ9Vδ2 T-lymphocytes, a small subset of T-cells that have important antitumor activity in bone marrow [44]. OS cells with low *ABCA1* are recognized with a lower efficiency by Vγ9Vδ2 T-lymphocytes [42], showing an immune-evasive attitude. Notably, doxorubicin-resistant OS cells have an increased rate of endogenous synthesis of FPP that activates Ras and determines an increased *ABCB1*/*ABCA1* ratio, which is translated into a chemoimmune-resistant phenotype [42]. This body of evidence provides a mechanistic lineage between cholesterol metabolism, ABC transporters, resistance to chemotherapy, and the host immune system. On the other hand, this phenotype offers an excellent Achilles’ heel in targeting the FPP synthase (FPPS), the enzyme that consumes IPP and produces FPP. Indeed, by inhibiting FPPS with the amino bisphosphonate zoledronic acid, Ras-dependent pathways can be slowed down, decreasing the *ABCB1*/*ABCA1* ratio. At the same time, IPP is accumulated and exported by *ABCA1* in the extracellular environment, where it efficiently recruits antitumor Vγ9Vδ2 T-lymphocytes [42]. Since the cholesterol biosynthetic pathway is highly druggable [45], these findings have unexpectedly paved the way for the use of cholesterol-modulating agents (statins, amino bisphosphonates) as adjuvants against chemoresistant OS.

The studies reported herein have been focused on the expression of ABC transporters in OS cells. Importantly, several ABC transporters of the A family are physiologically expressed on the immune-infiltrating cells, namely tumor-associated macrophages (TAMs), dendritic cells (DCs), and natural killer cells (NKs), which drive OS progression [21]. Here, ABCA transporters are physiologically involved in the export and exchange of phospholipids and cholesterol trafficking [39]. These functions support plasma membrane building, cell migration, and phagocytosis, thereby potentially enforcing the anti-OS activity of immune-active cells such as TAMs, DCs, and NKs [21]. Moreover, *ABCA1*, which is present in DCs, is the most active effluxer of IPP, which activates the expansion of Vγ9Vδ2 T-lymphocytes and their immune killing [43]; this mechanism is particularly effective against multiple myeloma [44]. We cannot exclude that the IPP effluxed by *ABCA1* present in DCs also induces the activation of Vγ9Vδ2 T-lymphocytes against OS cells. Similarly, the genetic deletion of *ABCA1* and *ABCG1* in TAMs slows ovarian cancer progression down by maintaining a low M2/M1 ratio [46]. Since a high M2/M1 ratio is associated with a more metastatic phenotype in OS [21], targeting of the ABC transporters effluxing cholesterol on immune-infiltrating cells could be a promising strategy to reduce OS progression, facilitating the efficacy of chemotherapy in eradicating the tumor or limiting the metastatic dissemination.

## 4. ABC Transporters and Drug Resistance in Ewing’s Sarcoma

The mechanisms by which ES becomes resistant to chemotherapy may involve different aspects, such as the expression of multiple drug resistance genes and drug efflux mechanisms, the presence of CSCs, the activation of proliferative intracellular pathways, or the acquisition of new mutations that allow tumor cells to escape the effects of chemotherapy [47].

In ES, the clinical role of ABC transporters and their association with patient outcomes still need to be clarified, although different studies have been performed on this issue.

The expression of *ABCB1* in ES has been assessed in different studies, although the reported data have often been discrepant.

In one of the first reported analyses of *ABCB1* in ES [48], no evidence of *ABCB1* gene expression was found in six ES untreated patients. In another similar study [49], *ABCB1* expression at the mRNA level was detected in two out of three untreated ES patients, although no evidence of *ABCB1* protein was revealed by immunohistochemistry.

One of the largest ES series that has been reported so far assessed *ABCB1* expression levels via immunohistochemistry in 87 primary non-metastatic tumors [50]. A positive immunoreaction for *ABCB1* was found in 27/87 samples (31%) and no evidence of association with clinical outcome was found in terms of event-free survival probability. These findings substantially confirmed the sporadic data that had previously been reported by other authors. Indeed, Roessner and co-workers had found a significant correlation between *ABCB1* protein expression and poor response to chemotherapy but with no clinical outcome in 21 ES tumor tissues [51], while two other studies had shown that *ABCB1* mRNA and protein expression were not predictive for ES prognosis [52,53]. Moreover, Hijazi and co-workers observed an increase in the number of ES samples expressing *ABCB1* after therapy, although a large proportion of these tumors already expressed this transporter in the absence of previous anticancer treatment [54]. Additionally, the same study did not reveal any significant differences between *ABCB1*-positive and -negative cases regarding relapse-free survival [54].

Additional data have been reported regarding other ABC transporters.

A trend toward poor overall survival was reported in ES patients with high expression of *ABCG2* [55].

A favorable prognosis for ES patients presenting high levels of *ABCA6* and *ABCA7* has been reported, although the role of these ABCA transporters in ES remains to be clarified [56].

Another transporter that has recently been studied in ES is the ATP-binding cassette subfamily F member 1 (*ABCF1*) [57]. *ABCF1* is a member of a subfamily of transporters composed of two nucleotide-binding domains, which do not possess membrane-spanning domains. *ABCF1* is known to be associated with eukaryotic initiation factor 2 and ribosomes, likely playing a role in mRNA translation [58,59]. In ES, it was found that *ABCF1* mRNA altered the functions of the insulin-like growth factor 2 mRNA-binding protein 3 (*IGF2BP3*) by serving as a sponge that limited *IGF2BP3* oncogenic potential [57]. The same authors also reported that *ABCF1* in association with *IGF2BP3* could predict ES outcome, since patients with high expression of *IGF2BP3* and low expression of *ABCF1* showed a worse prognosis [57].

ES research efforts have also focused on *ABCC1*.

*ABCC1* was found to increase drug efflux from mitochondria, inhibit the induction of mitochondrial-dependent apoptosis, and contribute to the development of drug resistance in ES cells [60]. *ABCC1* is glycosylated and localized to the outer mitochondrial membrane, where it is co-expressed with the chaperone *HSP90*, which in turn regulates the subcellular transportation of *ABCC1*, decreasing the response of ES cells to drugs that are *ABCC1* substrates [61].

Oda and co-workers detected *ABCC1* expression in eight out of ten ES tumor specimens, although only three of these samples expressed *ABCB1* at the mRNA level and no significant difference in the expression of *ABCC1* between untreated and treated tumors was found [62].

Functional splice variants of *ABCC1* have been described in normal and cancer cells, and some of these variants were also found in ES [63,64]. Among these, loss of *ABCC1* exon 9 was predictive for relapse in six ES patients, suggesting that alternative splicing may represent an additional mechanism of *ABCC1* expression control [53]; however, it must be underlined that the same study did not reveal any correlation between *ABCC1* expression level and chemotherapy-induced tumor necrosis, time to first relapse, or overall survival [53].

A role in ES drug resistance has also been described for the glioma-associated oncogene homolog 1 (*GLI1*), a transcription factor involved in sonic the hedgehog (*SHH*) signaling [65]. ABC transporters such as *ABCG2*, *ABCB1*, and ATP-binding cassette subfamily B member 2 (*ABCB2*) can increase drug resistance through an efflux activity induced by *GLI1* [66]. Moreover, *ABCB1* is a well-known target of *GLI1* [67], which can also contribute to vincristine resistance [68]. Indeed, it was recently demonstrated that vincristine-resistant ES cell lines exhibited upregulation of *GLI1* protein and *ABCB1* compared to their respective parental cell lines [68]. Moreover, *GLI1* downregulation in these vincristine-resistant ES cell lines proved to enhance sensitivity to vincristine in vitro, supporting the role of *GLI1* as a target to decrease vincristine resistance in ES cells [68]. Accordingly, targeting *GLI1* in ES cells with arsenic trioxide (ATO) was shown to inhibit the transcriptional activity of *GLI1* [69] and to suppress *GLI1*-dependent *ABCC1* and *ABCB1* transcription [70].

In conclusion, it can be stated that the body of evidence reported so far concerning the impact of ABC transporters on drug resistance or tumor progression in ES needs to be expanded in order to clarify whether these candidate biomarkers may be considered to modulate or innovate treatment strategies for this tumor.

## 5. ABC Transporters and Cancer Stem Cells in Osteosarcoma and Ewing’s Sarcoma

A consistent body of proof has shown that within cancer cell populations, a small subgroup of cells exists, termed CSCs, which retain peculiar characteristics and stemness markers. Such subpopulations have been identified in different solid tumors, including sarcomas [71], and have been shown to contribute to the vast heterogeneity of this class of neoplasms [72]. CSCs are postulated to reside at the apical positions of the cancer cell hierarchy, due to their dual ability to sustain cancer population growth and to replenish the CSC pool itself [73]. CSCs were described in sarcomas for the first time in 2005, when Gibbs and co-workers [74] isolated a small fraction of stemness-displaying cells from OS, while similar results were reported several years later for ES [75,76].

One of the methods used for the isolation of CSCs is based on the cytometric flow separation of tumor cells on the basis of their differential ability to efflux fluorescent dyes, such as Hoechst 33342 or rhodamine 123, a feature that is displayed by hematopoietic stem cells as well [77]. The dye extrusion ability observed in CSCs is bestowed by ABC transporters, such as *ABCG2* and *ABCB1*, which also confer drug resistance to these cells [78]. A small fraction of cancer cells with a stem-like phenotype that retain low amounts of Hoechst 33342 can be sorted and isolated. This subset of cells is termed the side population (SP), as flow cytometry analysis yields negatively stained cells confined on one side of the density dot plot. SP cells have been identified in several sarcoma cell lines and tumor specimens [75,79,80].

ABC transporters have been thoroughly investigated in OS CSCs, with an increasing number of studies published in the last ten years.

One of the first attempts to isolate the OS SP was carried out on seven OS cell lines, of which one was successfully sorted in SP (with high *ABCG2* expression) and non-SP cells [79]. Later, another group succeeded in isolating SP cells from OS tumor samples and performed microarray and molecular analyses, aiming to reveal genes that were differentially expressed in SP and non-SP cells; *ABCA2*, *ABCG2*, *ABCB1*, and *ABCC1* were found to be overexpressed in SP isolated from all OS samples [81].

Saini and co-workers (2012) compared *ABCB1*, *ABCC1*, and *ABCG2* protein levels between cells grown in either spherical or monolayer cultures and revealed that *ABCG2* was significantly increased in spherical cultures of all the three investigated OS cell lines [41]. As anticipated, transcriptome data detected higher *ABCA5* expression in spherical cultures compared to monolayer cultures of one OS cell line, while further investigation revealed a significantly higher expression of *ABCA5* in specimens from lung metastases compared to primary OS [41].

The analysis of one OS cell line with stemness properties selected from MG-63 cells by long term treatment with 3-aminobenzamide showed increased *ABCG2* expression compared to the parental cells [82]. The same group further studied this stem-like cell line and found 196 differentially expressed genes, including 11 ABC transporters: *ABCA5*, ATP-binding cassette subfamily A member 8 (*ABCA8*), *ABCB1*, ATP-binding cassette subfamily B member 10 (*ABCB10*), *ABCC1*, *ABCC4*, *ABCC5*, *ABCC6*, ATP-binding cassette subfamily D member 3 (*ABCD3*), and *ABCG2* [83].

Another study performed on cells cultured from untreated primary OS biopsies demonstrated the presence of SP cells with significantly enhanced chemoresistance towards cisplatin, doxorubicin, and methotrexate, which was associated with *ABCA1*, *ABCA2*, *ABCB1*, and *ABCG2* upregulation [84]. Additional evidence supporting the role of *ABCB1* and *ABCG2* in the enhanced chemoresistance of OS CSCs was reported by Gonçalves and co-workers [85].

In stem-like cells sorted from the MG-63 cell line, it was shown that downregulation of the DNA-dependent protein kinase catalytic subunit (DNA-PKcs) led to lower *ABCB1* expression and to cisplatin sensitization of CSCs, indicating another mechanism through which OS CSCs may become drug-resistant as a consequence of the increase in *ABCB1* [86].

In another study, SP cells isolated from the aggressive OS cell line OS-65 showed reduced sensitivity to DNA-damaging drugs, such as cisplatin, oxaliplatin, gemcitabine, paclitaxel, 5-fluorouracil, and etoposide, due to significantly increased expression of *ABCB1*, ATP-binding cassette subfamily B member 5 (*ABCB5*), and *ABCG2* as compared to non-SP cells [87].

Similarly, Roundhill and co-workers revealed increased expression of *ABCA1*, *ABCA9*, *ABCB1*, and ATP-binding cassette subfamily G member 1 (*ABCG1*) in self-renewing cells obtained from doxorubicin-resistant OS cells [88].

Finally, some studies have pointed out that stemness-associated pathways such as Notch homolog 1 (*Notch1*) [89] and *Wnt*/β-catenin [90,91] can affect the expression pattern of ABC transporters, and consequently the degree of resistance in OS CSCs, which are rich in *ABCB1*, *ABCB2*, *ABCG2* [92], *ABCC1* [89], or *ABCB5* [87].

Chemotherapy itself favors the expansion of CSC-enriched populations [88,93], enlarging the pattern of ABC transporters present in the surviving cells. This selection, as observed in OS preclinical models [88], phenocopies the expansion of resistant clones occurring in patients unresponsive to chemotherapy.

Differing from OS, very few findings have been reported concerning ABC transporters in ES CSCs.

The first successful isolation of SP cells in four out of five ES cell lines was reported in 2007 [75]. A few years later, Yang and co-workers succeeded in isolating an SP fraction of the ES cell line SK-ES-1 and showed that these SP cells displayed significantly enhanced clonogenicity, invasiveness, and resistance against cisplatin and doxorubicin due to overexpression of *ABCB1* and *ABCG2* [94].

It is, therefore, evident that additional studies and findings are needed to define the role and impact of ABC transporters in ES-derived CSCs.

## 6. ABC Transporters and Non-Coding RNAs in Osteosarcoma and Ewing’s Sarcoma

The discovery of non-coding RNAs (ncRNAs) opened the field for the identification of multiple mechanisms mediated by microRNAs (miRNAs) and long ncRNAs (lncRNAs), which control the expression of *ABCB1* and mediate drug resistance. Different miRNAs and lncRNAs have been found to be co-expressed in OS or to be easily transferred via extracellular vesicles to nearby cells, contributing to the simultaneous upregulation of *ABCB1* and other ABC transporter members in both autocrine and paracrine ways [95]. On the other hand, miRNAs downregulating *ABCB1* have also been documented in several solid tumors [96,97,98,99]; therefore, the knowledge of the ncRNA patterns in OS could pave the way toward new reversing strategies based on liposomes or nanoparticles carrying ABC-transporter-targeting miRNAs co-encapsulated with chemotherapeutic drugs that are usable in a broader range of malignancies [95].

Furthermore, ncRNAs have been shown to interfere with the expression of different ABC transporters in OS [95,100]. For example, the lncRNA ODRUL was found to be upregulated in doxorubicin-resistant OS cell lines, suggesting a crucial role of this lncRNA in doxorubicin resistance through the induction of *ABCB1* expression [101].

The effects of ncRNAs have also been documented in OS tissues from patients unresponsive to cisplatin, whereby the levels of *ABCB1* and the lncRNA ROR were simultaneously increased compared to responsive patients [102]. ROR was also aberrantly expressed in cisplatin-resistant OS cell lines, which showed simultaneously increased expression of *ABCB1*. Mechanistically, it was shown that ROR acts by sponging miR-153-3p, an miRNA that promotes the degradation of *ABCB1* mRNA, while ROR knockdown in cisplatin-resistant OS cells increased miR-153-3p and decreased *ABCB1* levels, restoring drug sensitivity [102].

Circular RNA (circRNA) is another emerging category of ncRNA mediating pleiotropic oncogenic functions. In recent years, several studies have reported the impacts of circRNAs on OS progression [103] and on modulation of ABC transporter expression.

For example, the circRNA PVT1 was found to be particularly higher in the serum of OS chemoresistant patients (being associated with poor prognosis), as well as in drug-resistant cell lines, in which it upregulated *ABCB1* [104]. Targeting PVT1 using specific siRNAs restored sensitivity to doxorubicin because of decreased *ABCB1* levels, indicating that this circRNA was able to influence the drug resistance of OS cells through the modulation of *ABCB1* [104].

An interesting observation with potentially important clinical impacts was derived from a study in which sirolimus, a first generation mTOR inhibitor, was found to induce apoptosis and reverses multidrug resistance in human OS cells by increasing miR-34b expression [105]. This study showed that treatment with sirolimus significantly sensitized the human OS cell line MG63/ADM to different anticancer drugs (doxorubicin, gemcitabine, and methotrexate), and also caused upregulation of miR-34b with a consequent apoptosis induction. Since miR-34b negatively modulates *ABCB1*, the authors suggested that miR-34b upregulation induced by sirolimus treatment may result in drug sensitization of the cells through the inhibition of *ABCB1*, in addition to the aforementioned apoptosis induction. This hypothesis was confirmed by the fact that miR-34b silencing reversed the sirolimus chemosensitizing effect in MG63/ADM cells. The authors further supported their in vitro findings with the results obtained in 40 tissue samples from OS patients, in which a significantly higher miR-34b and lower *ABCB1* immunohistochemical scores were detected in the subgroup of 17 chemosensitive patients, who also had a better outcome.

ABC transporters can also be indirectly modulated by genes affecting tumor malignancy, aggressiveness, and chemosensitivity. In OS cells, it was shown that the CCN family member 2 (*CCN2*) was able to induce *ABCG2* expression through different indirect mechanisms [20]. The first mechanism consisted of *ABCG2* upregulation as result of the binding of *CCN2* to α6β1 integrin and the subsequent downstream signaling modifications. The second mechanism was based on the *CCN2*-mediated decreased expression of miR-519d, a negative modulator of *ABCG2*. In agreement with these findings obtained in human OS experimental models, the same authors reported a positive correlation between *CCN2* and *ABCG2* protein expression as assessed by immunohistochemistry in a group of 40 OS patients, in which significantly higher *CCN2* expression was also found in the subgroup of drug-resistant patients [20]. Based on these results, the authors concluded that in OS, *CCN2* increases *ABCG2* expression and promotes drug resistance through the downregulation of miRNA-519d and that *CCN2* inhibition might be considered as a new therapeutic option for this tumor.

Differing from OS, no information about ncRNAs impacting on ABC transporters in ES has been reported so far.

## 7. Pharmacogenomics of ABC Transporters in Osteosarcoma and Ewing’s Sarcoma

In addition to the expression level, the presence of specific single-nucleotide polymorphisms (SNPs) in ABC transporters has also been associated with drug resistance in OS and ES patients.

Although many SNPs of ABC transporters have been studied in the germline DNA of OS patients using different candidate gene and high-throughput approaches, only a few of them have been reported to be associated with either response to treatment or survival, suggesting a possible involvement in patient drug resistance (Table 1).

The rs1128503 of *ABCB1*, a synonymous coding sequence SNP, is the most frequently studied SNP in patients with OS.

The TT genotype and the T allele were associated with good response to preoperative chemotherapy in three studies [106,107,108], although also with poor response in one study [109].

TT genotypes were associated with either better [107,108] or worse event-free and overall survival rates [106,109].

The C allele was associated with worse 5-year event-free and overall survival rates compared to the T allele [110].

These somehow contradictory results could be attributed to rather short median follow-up times and different ethnicities.

Caronia and co-workers reported the G allele of rs4148737 to be associated with worse and the C allele of rs1027636 with better 5-year survival rates; both of these alleles are intronic SNPs of the *ABCB1* gene [110]. They hypothesized that these SNPs could affect the efflux of drugs used in the treatment of OS. Furthermore, since rs1027636 was in complete linkage disequilibrium (LD) with rs1128503 but in low LD with rs4148737, they suggested that further functional studies were needed to elucidate which of the three SNPs of the *ABCB1* gene could be the causal variant.

The haplotype of the three frequently occurring SNPs of *ABCB1*, namely rs1128503, rs2032582, and rs1045642, was explored regarding their stability and substrate specificity in different experimental models [111]. Kimchi-Sarfaty and co-workers reported for the first time that the synonymous SNP rs1045642 was linked to altered substrate specificity and inhibitor interaction sites in HeLa cells [112]. Further studies in stable epithelial monolayers confirmed that this silent SNP changed the protein specificity and protein conformation, but did not influence the cell morphology, growth rate, or monolayer formation [113].

Although no functional studies of this common *ABCB1* haplotype have been reported so far in OS cells, or for its clinical relevance in patients with OS, a pharmacogenomic study carried out in renal transplant patients provided interesting information from a clinical setting [114]. The activity of the ABCB1 protein was lower in TT carriers of rs1128503, rs2032582, and rs1045642, supporting the functional consequences of these synonymous SNPs in vivo.

As far as the *ABCC2* gene is concerned, the CT/TT genotypes of rs717620, an SNP located in the 5′ untranslated region, were associated with a poor response to preoperative chemotherapy [115], whereas the AA/GA genotypes of the missense SNP rs2273697 were associated with worse event-free survival [116]. Pronounced inter-individual variability of *ABCC2* protein expression has been reported in most tumors [117] and functional studies of *ABCC2* haplotypes, including rs717620, rs2273697, and the synonymous SNP rs3740066, confirming that these polymorphisms influenced the transport capacity of *ABCC2* [118]. Furthermore, the study by Megaraj and co-workers 2011 showed that depending on their location, four non-synonymous SNPs of *ABCC2* altered the protein expression and substrate affinity [119]. For the rs2273697 variant, when mapping to the membrane spanning domain 1, a selectively decreased affinity for the glutathione and glucuronide substrates was reported. Since *ABCC2* was the third most expressed ABC transporter in human OS cell lines sensitive or resistant to doxorubicin, methotrexate, or cisplatin [12], and as all of these three drugs are substrates of *ABCC2*, it would be worthwhile conducting functional studies in OS cells in order to identify the causal mechanisms underlying these pharmacogenomic associations in patients with OS.

Consistent evidence for associations between the TT genotype and poor response to preoperative chemotherapy was reported for the synonymous SNP rs4148416 in exon 22 of *ABCC3* in two studies [107,109]. Associations between the T allele and worse 5-year event-free and overall survival compared to the CC genotype were first reported by Caronia and co-workers [110] and confirmed in two independent studies [107,109]. *ABCC3* conferred resistance to methotrexate in other cancers [120]; however, whether or not rs4148416 can influence the activity of the *ABCC3* protein has to our knowledge not been demonstrated so far.

Overall, the associations between the T allele of *ABCC3* rs4148416 with poor response and the T allele of *ABCB1* rs1128503 with good response, especially in Caucasian populations, were confirmed in a meta-analysis [121].

The G allele of the intronic rs939338 in the *ABCC5* gene was reported to be associated with worse 5-year event-free survival in two cohorts, including data obtained from germline (alive patients) and tumor samples (dead patients) [122]. This association was confirmed in a germline study including 132 OS patients [19]. Furthermore, in a subset of 52 patients, higher *ABCC5* RNA expression levels were revealed in tumor tissues of patients carrying the germline risk allele (GG/GA) compared to those with AA genotypes, showing that the elevated *ABCC5* expression might be associated with resistance to cisplatin and doxorubicin in OS patients [123].

The presence of single SNPs is not sufficient to determine a poor or good outcome. The coexistence of SNPs and altered expression in ABC transporters; drug-detoxifying enzymes such as glutathione S-transferase P1 (*GSTP1*) [107] or cytochrome P450 3A4 (*CYP3A4*) [122]; pro-apoptotic proteins such as Fas ligand (*FasL*) or caspase 3 (*CASP3*) [122]; and DNA repair machinery enzymes such as MutS homolog 2 (*MSH2*) [122] is part of a signature predictive of chemoresistance in OS, which is not entirely dependent on ABC transporters. In summary, although additional candidate SNPs of *ABCA1*, *ABCB1*, *ABCC1*, *ABCC2*, *ABCC4*, *ABCC6*, and *ABCG2* were analyzed in several studies [110,115,116,122], no associations with survival or response to therapy have been reported so far in patients with OS.

In ES, to explain the discrepancies reported for the role of polymorphisms affecting ABC transporters expression, it has been hypothesized that genetic variations in genes encoding for drug transporters, drug-metabolizing enzymes, or drug targets may be responsible for the variability in drug response exhibited by this tumor [124]. In particular, Ruiz-Pinto and co-workers performed a pharmacogenetic study genotyping 384 SNPs in 24 genes involved in the absorption, distribution, metabolism, and elimination of chemotherapeutic drugs used to treat ES [125]. Genetic variants of the ATP-binding cassette subfamily C member 6 (*ABCC6*) (rs7190447) and of the *ABCB1* gene (rs4148737) emerged as being significantly associated with overall survival in ES patients. The strongest evidence was found for the CC genotype of the intronic *ABCC6* rs7190447, which was found in only 1 patient in the discovery set (n = 97) and in two patients in the replication set (n = 305). The authors described that this SNP was in LD with the intronic rs7192303 located 122 bp upstream. They suggested that rs7192303 could be the most plausible causal SNP underlying the association with overall survival.

The GG genotype of rs4148737 in *ABCB1* was shown to be associated with higher risk of death, suggesting that this variant may be an important prognostic marker to be considered after treatment in ES. This intronic SNP is located in a weakly transcribed region and is the same one that was associated with decreased survival in pediatric patients with OS [110], suggesting that it might be of prognostic relevance in pediatric bone tumors. In addition, the authors reported that rs4148737 influenced expression levels in skeletal muscle, breast, and testis samples according to GTEx data. None of the associations between 20 germline variants and response to chemotherapy found in the discovery set (n = 77) were confirmed in the replication set (n = 197). In this analysis, several SNPs of *ABCC1*, *ABCC2*, *ABCC3*, and *ABCC4* were analyzed, without any relevant associations between genotype and therapy response or overall survival.

## 8. Treatment Strategies Targeting ABC Transporters in Bone Tumors

Blocking ABC transporters activity has been considered an attractive option to overcome MDR, also because many inhibitors have been identified and developed during the past decades [16,17]. Clinical trials aimed at improving the efficacy of chemotherapeutic drugs through inhibition of ABC transporters have been designed, although unfortunately they have invariably presented high levels of adverse toxicity, mainly due to the inhibition of the fundamental physiological detoxification activities exerted by these proteins [3,126]. This is the main reason explaining why ABC inhibitors have not successfully been marketed for clinical applications, in addition to the design of potent and selective agents with low toxicity still being a major challenge [127]; however, despite this serious limitation, the perspective of targeting *ABCB1* or other MDR-related ABC transporters with specific inhibitors or negative modulators is still considered a promising therapeutic strategy to overcome drug resistance in both OS and other tumors.

The previously provided information about the clinical relevance of *ABCB1* expression levels at diagnosis was considered to stratify OS patients and modulate treatment in the phase II–III Italian Sarcoma Group (ISG) trial ISG/OS-2 (https://ClinicalTrials.gov/show/NCT01459484, accessed on 28 May 2021) and in an observational trial of the Grupo Espanol de Investigacion en Sarcomas (GEIS) (https://ClinicalTrials.gov/show/NCT04383288, accessed on 28 May 2021), the results of which are presently under evaluation; however, it must be underlined that these trials used *ABCB1* protein expression as a stratification marker and did not employ any *ABCB1* targeting drug.

Recently, it has been indicated that inhibiting *ABCB1* transport activity with CBT-1^®^ (tetrandrine, NSC-77037), a member of a new generation of *ABCB1*/*ABCC1* inhibitors, may open up novel therapeutic opportunities for patients with *ABCB1*-overexpressing OS. Indeed, targeting *ABCB1*/*ABCC1* with CBT-1^®^ in drug-resistant human OS cell lines restored sensitivity to doxorubicin and also to drugs used in OS second-line therapy, such as etoposide, taxotere, and vinorelbine [12].

CBT-1^®^ recently entered clinical trials in combination with different chemotherapeutic drugs and proved to negatively interfere with ABC transporters at tolerable doses, without affecting the dosages of conventional agents [128,129]. Based on the aforementioned findings, the use of CBT-1^®^ in association with conventional chemotherapeutic drugs has been indicated as a new therapeutic option for refractory or recurrent OS patients, and has been included in a phase I–II clinical trial recruiting metastatic, unresectable sarcoma (including OS) patients who have progressed after treatment with doxorubicin (https://ClinicalTrials.gov/show/NCT03002805, accessed on 28 May 2021).

Another strategy that was taken into consideration to inhibit *ABCB1* activity in human tumor cells was based on natural products such as curcumin, which was included in a phase I–II clinical trial for relapsed or metastatic high-grade OS patients (https://clinicaltrials.gov/show/NCT00689195, accessed on 28 May 2021); unfortunately, this trial has not provided results yet, despite being officially closed in 2013. Curcumin is a phenolic compound used in traditional Indian and Asian medicine, which have been proven to inhibit expression and function of several ABC transporters [130] and to partially revert the *ABCB1*-mediated doxorubicin resistance in human OS cell lines, although with remarkably lower efficiency compared to conventional *ABCB1* inhibitors [12].

Thanks to the increasing knowledge of molecular mechanisms regulating the expression of *ABCB1*, treatment approaches based on the combination of first-line chemotherapy with agents downregulating *ABCB1* by targeting upstream signaling pathways have been developed.

For instance, zoledronic acid, which downregulates *ABCB1* in OS cells [42], has been investigated regarding whether it may improve the efficacy of chemotherapy in OS. After a dose escalation study that established the maximum tolerated dose of zoledronic acid co-administered with first-line chemotherapy in patients younger than 40 years with metastatic OS (https://ClinicalTrials.gov/show/NCT00742924, accessed on 28 May 2021) [131], phase II (https://ClinicalTrials.gov/show/NCT00691236, accessed on 28 May 2021) and phase III trials (https://ClinicalTrials.gov/show/NCT00470223, accessed on 28 May 2021) were launched. In these trials, OS patients were treated with chemotherapy alone or with chemotherapy plus zoledronic acid, respectively. Although no results have been posted for the first clinical study (NCT00691236), the second trial (NCT00470223) demonstrated that the addition of zoledronic acid to conventional chemotherapy did not improve treatment efficacy or patient prognosis [132]; however, it can be speculated how zoledronic acid could be used inside subgroups of *ABCB1*-overexpressing patients.

In ES, based on the results obtained in glioblastoma, in which *ABCB1* expression and activity have been downregulated by temozolomide [133,134], trials combining temozolomide with standard chemotherapy were designed in order to exploit the cytotoxic peculiar mechanisms of temozolomide together with the temozolomide-induced inhibition of *ABCB1*. Unfortunately, a phase II trial (https://ClinicalTrials.gov/show/NCT01313884, accessed on 28 May 2021), which compared the standard chemotherapy regimen versus the addition of temozolomide in patients with metastatic ES, did not reach neither its primary objective (partial or complete response), nor its secondary objective (disease-free survival), because it was not able to recruit the number of patients required to reach sufficient statistical power; however, a second phase II trial (https://ClinicalTrials.gov/show/NCT03359005, accessed on 28 May 2021) comparing the efficacy of second-line chemotherapy (vincristine and irinotecan) versus chemotherapy plus temozolomide in relapsed and metastatic ES patients is presently recruiting patients and is expected to provide useful information about the feasibility and efficacy of this therapeutic approach.

Additional treatment strategies that can interfere with *ABCB1* activity have been indicated by recent studies on protein kinase inhibitor drugs. These inhibitors have been suggested as promising agents to improve the efficacy of conventional treatments in several human cancers and many of them are presently also under investigation in OS [135,136]. Despite the resistance against several kinase inhibitors that can occur from the upregulation of *ABCB1* or other ABC transporters [137,138], there is also evidence that some of these drugs can produce downregulation of ABC transporter activity, acting as chemoresistance revertants [138,139] or partially overcoming drug resistance when used in association with other ABC substrates. The latter situation was described in OS experimental models, in which combined treatment of the Aurora kinases inhibitor VX-680 (MK-0457, tozasertib) with doxorubicin proved to overcome drug resistance in *ABCB1*-overexpressing human OS cell lines [140], most probably because of the simultaneous presence of two *ABCB1* substrates, with a consequent impairment of its efflux activity.

New possible targets and strategies of intervention to overcome drug resistance in OS have been highlighted by experimental studies investigating factors and pathways that can influence ABC transporter expression and activity.

The estrogen-related receptor α (*ERRα*) was recently pointed out as an important controller of *ABCB1*, because it is a direct transcriptional inducer of the *ABCB1* gene, while at the same time it inhibits miR-9, which destabilizes the *ABCB1* mRNA. This circuit can be targeted pharmacologically, since *ERRα* is druggable. Indeed, the *ERRα* inhibitor XCT-790 was demonstrated to overcome resistance to doxorubicin and cisplatin in chemoresitant OS cells [141].

Peroxisome-proliferator-activated receptor gamma (*PPARγ*) is another factor involved in the transcription of *ABCB1* in patient-derived orthotopic xenografts (PDOXs) resistant to doxorubicin, where it synergizes with β-catenin, a strong *ABCB1* inducer. Regarding *ERRα*, this observation also has translational potential because the antidiabetic drug pioglitazone, a *PPARγ* agonist, was shown to disrupt the *PPARγ*/β-catenin synergism, downregulating *ABCB1* and reversing doxorubicin resistance in PDOXs [142].

Oncogenic pathways converging on pro-survival factors, such as nuclear factor-kB (*NF-kB*) and activator protein 1 (*AP-1*), are other inducers of *ABCB1* in OS. The *NF-kB*/*AP-1*-driven transcription of *ABCB1* is antagonized by the tumor necrosis factor alpha (*TNF-α*)-induced protein 8-like 2 (*TIPE2*), whose overexpression was proven to sensitize OS xenografts to cisplatin [143]; therefore, agents targeting *TNF-α* or *NF-kB* inhibitors may be considered as candidate downregulators of *ABCB1* and potential chemosensitizer agents in OS.

*HIF-1α*, which is typically activated in the hypoxic niche of OS, is another inducer of the *MDR1* gene [144], which can also be targeted by different specific inhibitors, some of which are under clinical evaluation in other tumors.

Finally, epigenetic events may contribute to upregulate *ABCB1* via histone deacetylase 6 (*HDAC6*), which is significantly more expressed in doxorubicin-resistant OS cells than in sensitive ones. *HDAC6* upregulates interleukin-8 (*IL-8*), which in turn increases the transcription of the *ABCB1* gene in an autocrine way. This axis offers new therapeutic opportunities for high-grade OS patients, because HDAC inhibitors or agents targeting *IL-8* in the tumor microenvironment could be exploited as doxorubicin-sensitizing agents [145].

## 9. Conclusions

As summarized in Appendix A, in addition to their role as major determinants of chemoresistance in OS and ES, ABC transporters have been shown to influence cellular processes closely associated with the malignant potential of cancer cells, including proliferation, differentiation, migration, and invasion, through drug-efflux-independent functions [15,146].

Although extensive evidence has supported the role of ABC transporters in the drug resistance of OS, in ES, it is necessary to hypothesize a more general role that goes beyond chemoresistance.

Efforts aimed at reversing MDR in cancer cells have led to the discovery of a broad range of ABC transporter inhibitors, most of which are unfortunately not usable in clinical trials because of their unacceptable adverse toxicity; therefore, further optimization and synthesis of ABC transporter inhibitors is highly warranted to improve their specificity and safety. Moreover, to identify patients eligible for treatment with these new agents, ABC transporter expression must be reliably detected in tumor tissue, while clinically validated methods for detection of ABC protein expression must be standardized.

## Figures and Tables

**Figure 1 cells-10-02461-f001:**
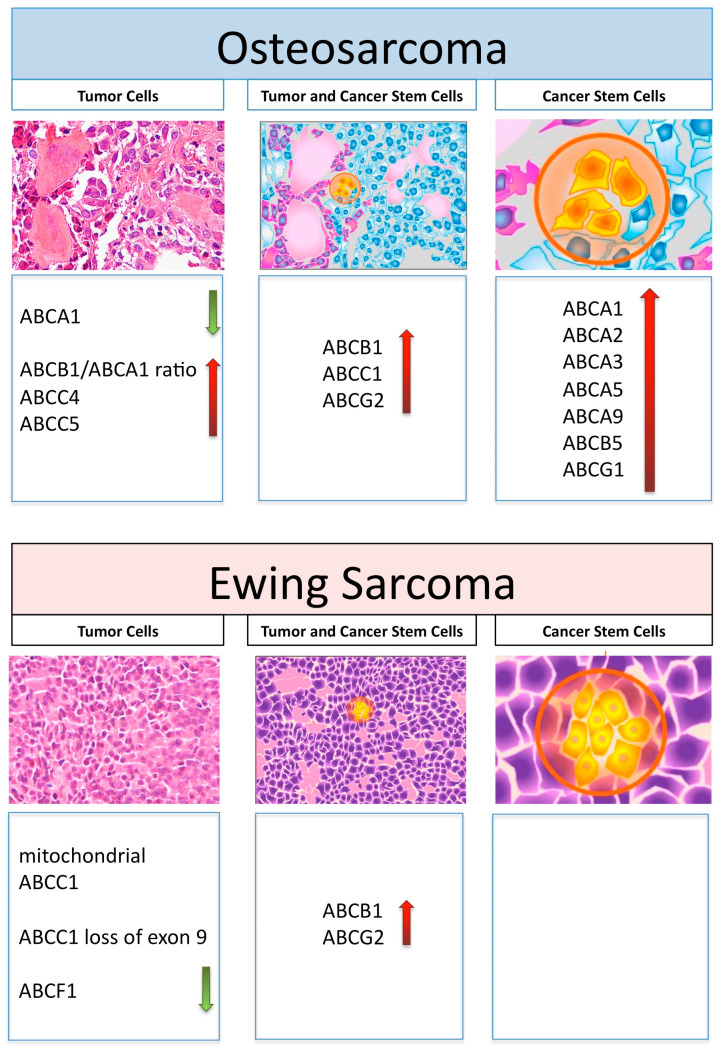
Graphical representation of histology and ABC transporters, which have been reported to be involved in drug resistance in tumor cells or cancer stem cells and in both cell types in osteosarcoma and Ewing’s sarcoma. Increased and decreased expression levels are shown by red and green arrows, respectively.

**Table 1 cells-10-02461-t001:** Germline single-nucleotide polymorphisms of ABC transporters reported to be associated with responses to preoperative chemotherapy or survival in patients with osteosarcoma or Ewing’s sarcoma.

Gene	rs Number	Genotype	Function (Effect on Protein)	Clinical Impact
**Osteosarcoma**				
*ABCB1* *	rs1128503	TT	cds-synon (Gly412Gly)	good or poor Resp, decreased or increased EFS and OVS
*ABCB1*	rs4148737	G allele	intronic	decreased 5-year survival
*ABCB1*	rs1027636	C allele	intronic	increased 5-year survival
*ABCC2*	rs717620	CT/TT	5′ untranslated region	poor Resp
*ABCC2*	rs2273697	AA/GA	missense (Val417Ile)	decreased EFS
*ABCC3* *	rs4148416	T allele	cds-synon (Gly1013Gly)	poor Resp, decreased 5-year EFS and OVS
*ABCC5*	rs939338	G	intronic	decreased 5-year EFS
**Ewing Sarcoma**				
*ABCB1*	rs4148737	G	intronic	decreased 5-year OVS
*ABCC6*	rs7190447	CC GG/GC	intronic	higher risk of death increased 5-year OVS

Legend: cds-synon, coding sequence—synonymous; EFS, event-free survival; OVS, overall survival; Resp, response to preoperative chemotherapy. * Indicates associations confirmed in meta-analyses.

## Data Availability

Not applicable.

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
