# Peer review of "Impact of ABC Transporters in Osteosarcoma and Ewing’s Sarcoma: Which Are Involved in Chemoresistance and Which Are Not?"

_cells, 2021, doi:10.3390/cells10092461_

Round 1
Reviewer 1 Report
In this manuscript, the authors summarise the knowledge about the role of ABC-transporters on the chemoresistance of osteosarcomas and Ewing sarcomas.
This review article is written in excellent English, but I found it very difficult to follow and of limited value for those interested in the field, since the presentation of the information is not well organised and the authors generally do not evaluate and discuss the evidence. I feel that a more concise and critical review, bridging the preclinical and clinical findings, would be useful. I also find Figure 1 not very informative, would suggest a table where each ABC transporter is evaluated regarding the features presented, with the relevant references shown.
Reviewer 2 Report
There are many reasons why ABC transports can be regulated and, importantly, increased expression of an ABC transporter in a tumor does not necessarily translate into MDR. Too often the authors use the fact that a certain ABC protein is upregulated as kind of proof that the protein is involved in MDR.
Although many papers have described overexpression of ABC transporters in cancer specimen, most did not discriminate between overexpression in tumor cells and overexpression in immune cells infiltrating the tumor. The authors should discuss this important point.
English needs attention, I leave this to Editorial office.
Reviewer 3 Report
The manuscript by Serra and colleagues is a review of literature about the role of ABC transporters in two bone sarcomas, osteosarcoma and Ewing’s sarcoma, and their involvement in chemoresistance. ABC transporters is a family of proteins well known to be a major actor of drug resistance in a large part of cancers. They will allow chemotherapeutic drugs to be expelled from the tumoral cells. Because of this ability ABC transporters are responsible of multidrug resistance (MDR) phenotype and are good candidate therapeutic targets.
Comments
- Introduction
Figure 1: This figure is difficult to understand. It is not clear what the circles and histology images represent. The legend or the text are incomplete and do not provide any precision. The figure seems related with chapter 5 but not with the introduction.
- ABC transporters and drug resistance in osteosarcoma
“ABCB1 increased expression has also 73 been shown to be more frequent in pretreated high-grade OS compared with untreated 74 tumors and to be linked to a higher degree of metastasis formation”. Could the authors give an explanation or a hypothesis about the pro-metastatic effect of ABCB1?
- ABC transporters not involved in chemoresistance in osteosarcoma
A large part of this chapter is about ABCB1/ABCA1 ratio and chemoresistance. This seems to be in contradiction with the title of the chapter. I suggest that the authors reorganize this part or rename it.
- ABC transporters and drug resistance in Ewing’s sarcoma
“some ABC transporters can interact with other molecules by affecting 183 tumor malignancy, aggressiveness and chemosensitivity, as was shown for CCN family 184 member 2 (CCN2) which regulates ABCG2 in OS” Could the authors be more specific about ABCG2 regulation by CCN2? What is the consequence?
- ABC and cancer stem cells in osteosarcoma and Ewing’s sarcoma
The word “transporters” is missing in the title.
- ABC and non-coding RNAs in osteosarcoma and Ewing’s sarcoma
The word “transporters” is missing in the title.
“An interesting observation with important potential clinical impact derived from a 335 study in which sirolimus, a first generation mTOR inhibitor, was found to induce apop-336 tosis and reverses multidrug resistance in human OS cells via increasing miR-34b ex-337 pression” The authors should specify that miR-34b seems targeting ABCB1 mRNA.
- Pharmacogenomics of ABC transporters in osteosarcoma and Ewing's sarcoma
It will be interesting to describe briefly how the different SNPs affect ABC transporters functions.
The manuscript submitted by Serra and colleagues is clear, informative and satisfies the required quality criteria. If authors are willing to answer to my questions/requests, I will be then favorable for acceptance.
Round 2
Reviewer 1 Report
The revised version of the manuscript is clearly improved. I would highly recommend a Table briefly summarising the available data for each ABC-transporter as a supplementary material, since the text not very concise.
Author Response
Re: “Impact of ABC transporters in osteosarcoma and Ewing's sarcoma: which are involved in chemoresistance and which are not?” (Manuscript ID: cells-1300589) for the Special Issue “Molecular and Cellular Mechanisms of Cancers: Bone Sarcomas - Series 2”.
Authors reply to Reviewer 1
According to the Reviewer's suggestion, in the revised manuscript we have added a Supplementary Table, in which the roles of the considered ABC transporters have been summarized.
We would like to thank the Reviewer for this valuable suggestion, which surely improved the quality of our review.
Sincerely,
Dr. Massimo Serra
Corresponding Author on behalf of all co-Authors
IRCCS Istituto Ortopedico Rizzoli
Via di Barbiano, 1/10 - 40136 Bologna - Italy
TEL (+39) 051-636 6762 - e-mail: massimo.serra@ior.it
Reviewer 2 Report
English language and style are fine/minor spell check required
Author Response
English language and spelling has been further checked by a native English-speaking colleague (Pr. Joanna Kopecka), who has now been acknowledged.
Changes in the text have been highlighted in BLU.
We would like to thank the Reviewers for their valuable comments and remarks, which we have addressed in this revised version.